# SwinZS3: Zero-Shot Semantic Segmentation with a Swin Transformer

## Abstract

Zero-shot semantic segmentation (ZS3) aims at learning to classify the never-seen classes with zero training samples. Convolutional neural networks (CNNs) have recently achieved great success in this task. However, their limited attention ability constraints existing network architectures to reason based on word embeddings. In this light of the recent successes achieved by Swin Transformers, we propose SwinZS3, a new framework exploiting the visual embeddings and semantic embeddings on joint embedding space. The SwinZS3 combines a transformer image encoder with a language encoder. The image encoder is trained by pixel-text score maps using the dense language-guided semantic prototypes which are computed by the language encoder. This allows the SwinZS3 could recognize the unseen classes at test time without retraining. We experiment with our method on the ZS3 standard benchmarks (PASCAL VOC and PASCAL Context) and the results demonstrate the effectiveness of our method by showing the state-of-art performance.

## 1 Introduction

Semantic segmentation is at the foundation of several high-level computer vision applications such as autonomous driving, medical imaging, and so on. Recent deep learning has achieved great success in semantic segmentation Chen et al. (2018); Long et al. (2015); Ronneberger et al. (2015); Zhao et al. (2017b). However, the fully-supervised semantic segmentation models usually require extensive collections of labeled images with pixel-level annotations. And it could only handle the pre-defined classes. Considering the high cost of collecting dense labels, recently weakly supervised semantic(WSSS) segmentation methods have been explored. WSSS are often based on easily obtaining annotations, such as scribbles Sun et al.,bounding boxes Dai et al. (2015), and image-level labels Hou et al. (2018). Among them, a popular trend is based on network visualization techniques like classification activation map generating pseudo ground-truths Zeiler & Fergus (2014); Zhang et al. (2020). However, these methods also require the networks to have labeled images.

On the contrary, humans could recognize novel classes with only descriptions. With the inspiration of this, some recent methods seek zero-shot semantic segmentation (ZS3) Zhao et al. (2017a); Bucher et al. (2019); Gu et al. (2020); Li et al. (2020) . ZS3 benefits from semantic level supervision from texts by exploiting the semantic relationships between the pixels and the associated texts, which makes it enjoy a cheaper source of training data. The ZS3 methods could be categorized into generative and discriminative methods Baek et al. (2021). Both could predict unseen classes using only language-guided semantic information of the corresponding classes. For the generative ZS3 methods Creswell et al. (2018); Kingma & Welling (2013), segmentation networks are first trained with only seen classes labeled data. Then, they freeze the feature extractor to extract seen classes' visual features and train a semantic generator network to translate the language embedding to visual space. By doing this, the semantic generator could generate visual features conditioned on language embedding vectors. Finally, a classifier is trained to classify the features combined with features produced by the feature extractor on seen classes and generated features produced by the semantic generator from language embeddings on unseen classes. With generative methods achieve impressive performance in zero-shot semantic segmentation tasks. The methods are limited by a multi-stage training strategy, and the visual features extracted from the feature extractor not considering the language information during training. This will cause a seen bias problem towards the visual and generated features.

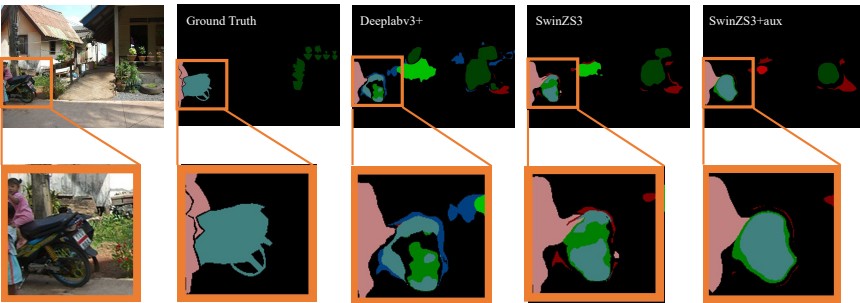

Figure 1: Effects of transformer's global reason and the score map decision boundary for zero-shot semantic segmentation. The motorbike(blue) is the unseen class. The existing solutions Deeplabv3+ often yield inaccurate segmentation results with limited receptive field and attention ability, losing fine-grained details. Using the transformer extractor significantly improves the prediction accuracy of the unseen classes. But there is still seen bias problem, which classifies the unseen class pixels into seen classes. So, our SwinZS3 proposes a language-guided score map to reduce it

To overcome the limitation, we introduce a discriminative approach for ZS3 that exploits to train visual and language encoders on a joint embedding space. During the training time, we avoid the multi-stage training strategy. We alleviate the seen and unseen bias problem by minimizing the euclidean distances and using the pixel-text score maps between the semantic prototypes produced by the language encoder and the visual features of corresponding classes. At test time, we use the learning discriminative language prototypes and combine the pixel-text score map with the euclidean distance as the decision boundary to avoid retraining.

Improving the network backbone of zero-shot networks is another effective way to reduce the bias problem. As shown in Figure.1 We argue that a shared shortcoming of previous ZS3 models falls in the reduced receptive field of CNNs and less using attention mechanisms for extracting the global relations of visual features conditioned with language semantic information. The local nature of convolutions leads CNNs to extract visual features missing long-range relationships across the same image. CNNs based frameworks sometimes fail to extract language-guided activation fields for lacking the global perceiving attention mechanism. Recently, transformers Vaswani et al. (2017) have significant breakthroughs in both of natural language processing(NLP) and computer vision(CV) field Xie et al. (2021a); Zheng et al. (2021); Arnab et al. (2021). Dosovitskiy et al. (2020) (ViT) is the first work to apply the transformer architecture to image classification. Moreover, the Liu et al. (2021) (swin-transformer) presents a new architecture for more general-purpose vision tasks, especially dense predicting. We argue that the self-attention mechanism benefits the zero-shot semantic segmentation tasks and the semantic information supervisor. The transformer-based model could capture the global feature relations and the semantic information in visual features by multi-head self-attention(MHSA).

This paper takes this missing step and explores the swin-transformer for ZS3. It combines convolutional layers and transformer blocks to model the global information guided by pixel-text distances and score maps. We also improve the decision boundary by modifying the Nearest Neighbor(NN) Classifier with weighted euclidean distance by score map. We demonstrate the effectiveness of our approach on standard zero-shot semantic segmentation benchmarks, achieving state-of-the-art performance on PASCAL-VOC Everingham et al. (2010) and PASCAL-Context Mottaghi et al. (2014).

Some methods based on CLIP (Radford et al. (2021)Xu et al. (2022)Ding et al. (2022)Arnab et al. (2021) Xu et al. (2021)) often claim to be zero-shot learning methods. However, those methods usually use all classes of images and text labels during training, which will cause supervision leakage.

## 2  RELATED WORK

**Semantic segmentation:** Semantic segmentation has made great advancements due to the rise of deep learning. Most recent state-of-the-art models are based on fully convolutional neural networks Long et al. (2015) and assume that all the training data have pixel-level annotations. DeepLab

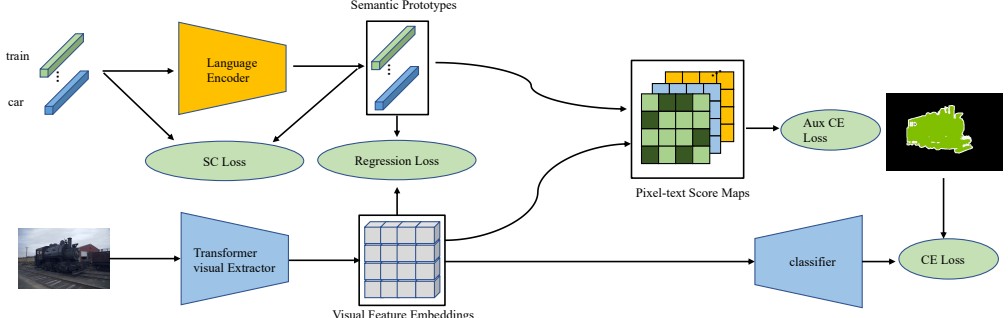

Figure 2: The overall framework of our approach SwinZS3. SwinZS3 first extracts the image visual embeddings using a transformer-based feature extractor and $K$-class semantic prototypes using a language encoder. The prototypes will have a regression loss with the visual features and their inter-relationship are transferred from the language embeddings ($word2vec$) using semantic consistency loss. And then SwinZS3 calculates pixel-text score maps in a hyper-sphere space for the projected visual features and semantic prototypes. The score maps are supervised by the ground-truth labels. The visual features are also fed into a classifier for being supervised by ground-truth labels with cross-entropy loss.

exhibited distinct performance improvement on the PASCAL VOC2012 Everingham et al. (2010) and MS-COCO Lin et al. (2014) by using the multiple scales Li et al. (2020); Zhang et al. (2018) and dilated convolution Chen et al. (2014; 2017). Other methods like UNet Ronneberger et al. (2015) and SegNet Badrinarayanan et al. (2017) also achieve an impressing performance by different strategies. The ViT Dosovitskiy et al. (2020) is the first work to use transformer architecture for recognition task. And swin transformer extends transformer to the dense prediction task and achieved the state-of-the-art performance. But they all heavily rely on expensive pixel-level segmentation labels, and they assume that training data for all categories is available beforehand. Recently, lots of weakly supervised semantic segmentation (WSSS) methods which are based on easily obtained annotations such as bounding boxes Dai et al. (2015), scribbles Sun et al. and image-level labels Hou et al. (2018) have been explored. The key to prevailing pipeline WSSS is to generate the pseudo-labels, especially network visualization techniques like CAM. And some works use growing strategies to grow the CAM ground-truth regions to the entire objects . But it is still difficult to get the pseudo-labels revealing entire object areas with accurate boundaries for the ill-posed procedure Singh & Lee (2017); Li et al. (2018).

**Zero-shot semantic segmentation**: The ZS3 networks could be categorized into discriminative and generative methods. For the discriminative approach, the work of Zhao et al. (2017a) focuses on hierarchical predicting unseen classes by adopting the discriminative approach. SPNet Xian et al. (2019) exploits a semantic embedding space by mapping visual features to fixed semantic ones. JoEm Baek et al. (2021) propose to align visual and semantic featuers in joint embedding space. In contrast to discriminative methods, ZS3Net Bucher et al. (2019) synthesize visual features by a generative moment matching network (GMMN). However, the ZS3Net training pipeline consists of three stages that will cause the bias problem. CSRL Li et al. (2020) exploit the relations of both seen and unseen classes to preserve them to synthesized visual features. CaGNet Gu et al. (2020) proposes to use the channel-wise attention mechanism in dilated convolutional layers for extracting visual features.

**Visual-language learning**: Recently years, image-language pairs learning is a rapidly growing field. There are some representative works such as CLIP Radford et al. (2021) and ALIGN Jia et al. (2021) which are pretrained on hundreds of millions of image-language pairs. Yang et al. (2022) presents the unified contrastive learning method that can leverage both the image-language methods and image-label data. And our work further extends the method to pixel-level on ZS3.

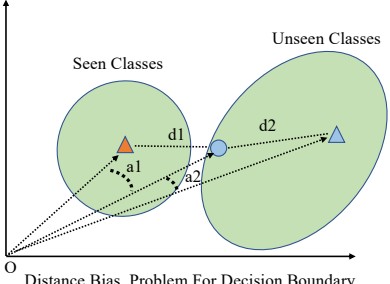 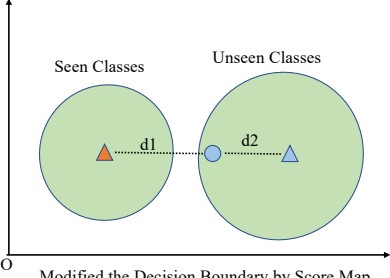

Figure 3: Compare the difference between the decision boundary without and with the score map. We visualize the visual features by circles and the semantic prototypes by triangles. Because of the bias problem in zero-shot learning, the visual features of seen classes are tightly, and the unseen classes' semantic prototypes and visual features are biased. (a) We show one of the situations where the euclidean distance $d1$ is smaller than $d2$. So, the unseen classes' pixels will be classified into seen classes. However, for score map distance show $a1$ is bigger than $a2$, which inspires us to use the score map distance to modify the euclidean distance. After adjusting, we could get the (b) view. It is crucial to improve the performance in ZS3.

## 3 METHOD

### 3.1 MOTIVATIONS

Unlike supervised semantic segmentation methods, The unseen classes prototypes of discriminative zero-shot semantic segmentation rely on joint optimization of the visual encoder and language encoder. Thus, to achieve good performance, this formulation requires the network to perceive the language context structure. Current network Baek et al. (2021) adopt traditional convolutional layers for aggregating language information. However, the intrinsic locality and weak attention of the convolution operator can hardly model long-range and accurate visual-language joint features. So, we propose to use transformer-based blocks to address the limitation. Another limitation of ZS3 is the seen bias problem. Lacking the unseen classes labeled data, it is difficult for the visual encoder to extract distinguishable features. As shown in Fig.3, modulating the decision boundary could also reduce the bias problem. We analyze the shortcomings of traditional NN classifiers and use the new decision boundary to improve the performance. Driven by this, we design our SwinZS3 framework. As shown in Fig.2.

### 3.2 OVERVIEW

Following the common practice, we divide classes into seen classes $S$ and unseen classes $U$. During the training time, we train our model, which includes a visual feature extractor and a semantic prototype encoder with the seen classes set $S$ only. Zero-shot semantic segmentation aims to allow the model to recognize both seen classes $S$ and unseen classes $U$ in the test time. We use the visual extractor to extract visual features and input language embeddings ($word2vec$) to the language encoder for obtaining semantic prototypes of corresponding classes. The visual features will be input into a classifier and supervised by ground-truth labels. The language prototypes will have a regression loss with the visual features. And the prototypes' inter-relationship are transferred from the language embeddings like $word2vec$ using the semantic-consistency loss. And then SwinZS3 calculates pixel-text score maps in a hyper-sphere space for the projected visual features and semantic prototypes. These score maps are also supervised by the ground-truth labels. The visual features are also fed into a classifier for being supervised by ground truth labels. In the following, we describe our framework in detail.

### 3.3 TRANSFORMER BACKBONE

Our framework uses the transformer Liu et al. (2021) as backbone that an input image is split into non-overlapping $h \times w$ patches. The patches will be projected to $h \times w$ tokens. The multi-head self-attention (MHSA) layer is used for the transformer block to capture the global features information. In MHSA, the patch tokens are projected to queries $Q \in \mathbb{R}^{hw \times d_k}$, keys $K \in \mathbb{R}^{hw \times d_k}$, and values $V \in \mathbb{R}^{hw \times d_v}$. $h$ and $w$ are the shapes of feature maps. $d_k$ and $d_v$ denote the dimension of features. Based on $Q$, $K$, and $V$, outputs X are

$$X = softmax(\frac{QK^T}{\sqrt{d}})V \tag{1}$$

The MHSA is the core operation of the transformer block. The transformer backbone's final output is produced by stacking multiple transformer blocks.

### 3.4 NETWORK TRAINING

As shown in Fig.2, our framework consists of four loss terms : cross-entropy loss $L_{ce}$ Misra & Maaten (2020), pixel-wise regression loss $L_r$, pixel-text score map loss $L_{aux}$, semantic consistency loss $L_{sc}$. The overall loss is finally formulated as

$$L = L_{ce} + L_r + \lambda_1 L_{sc} + \lambda_2 L_{aux} \tag{2}$$

where $\lambda_1$ and $\lambda_2$ balance the contributions of different losses.

**Cross-entropy loss:** Given the final outputs of feature maps $\upsilon \in \mathbb{R}^{h \times w \times c}$, where $h$, $w$, and $c$ are the height, width, and the number of channels. Then, $\upsilon$ will be put into a classifier head $f_c$. For the zero-shot setting, the classifier could learn the seen classes. So, we apply the cross-entropy loss Murphy (2012) that is widely adopted in supervised semantic segmentation on the seen classes set $S$ as follows:

$$L_{ce} = -\frac{1}{\sum_{c \in S} | N_c |} \sum_{c \in S} \sum_{p \in N_c} log(\frac{e^{w_c \upsilon(p)}}{\sum_{j \in S} e^{w_j(\upsilon(p))}}) \tag{3}$$

where $N_c$ indicates a set of locations labeled as the class $c$ in ground-truth.

**Regression loss:** Although the $l_{ce}$ could train the model to a discriminative embedding space on seen classes $S$. However, the model is not adaptable to classify the unseen classes $U$ while the classifier head does not learn the unseen classes prototypes. At test time, we want to use the language prototypes of both seen and unseen classes as the classifier to recognize the dense visual features extracted by the transformer backbone. For this, the distances of visual features and corresponding language prototypes should be minimized in embedding space. To address it, we introduced regression loss $l_r$. As the $l_{ce}$, we first get the final outputs visual feature maps $\upsilon \in \mathbb{R}^{h \times w \times c}$. Then, we get semantic feature maps $s \in \mathbb{R}^{h \times w \times d}$ where each pixel $s_c$ of $s$ is a word or language embedding and the same class with corresponding visual feature pixel. Given the language embedding maps, we input them to semantic encoder $f_s$ as follows:

$$\mu = f_s(s) \tag{4}$$

where $\mu \in \mathbb{R}^{h \times w \times c}$. We denote each pixel of $\mu_c$ is a semantic prototype for a class $c$. So, the regression loss as follows:

$$L_r = \frac{1}{\sum_{c \in S} | R_c |} \sum_{c \in S} \sum_{s \in R_c} d(\upsilon(s), \mu(s)) \tag{5}$$

where $d()$ is the euclidean distance metric. $R_c$ means the regions labeled with the same class in the ground truth. The $l_r$ give a promise that the dense visual features and semantic prototypes will be projected to a join embedding space where the pixels of corresponding classes will be close. But, for ZS3, there are similar limitations with $l_{ce}$: The $l_r$ deal with pixel-wise visual features and semantic

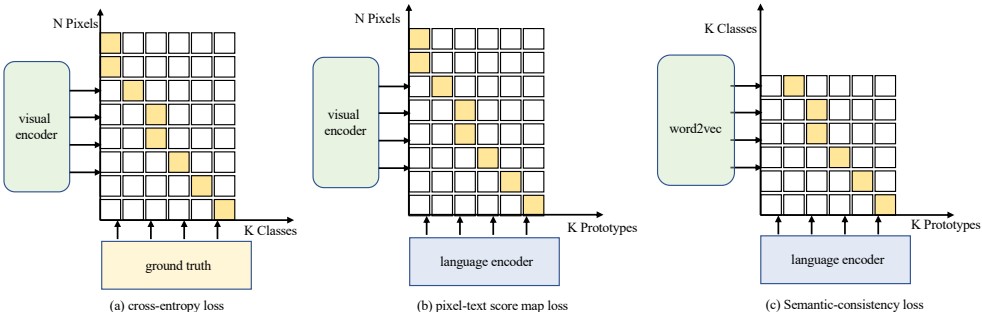

Figure 4: An illustrative similarity matrix comparisons between different losses : N is the number of pixel samples fro image features extracted by visual encoder. K is the number of classes. (a) The cross-entropy loss could be considered as pixel-label learning by assigning pixels to ground-truth labels, and the relationship between the pixels and labels is many-to-one. (b) The pixel-text score map loss focuses on the relationship for the semantic prototypes and the visual features, which means a semantic prototype will be assigned to many pixels. (c) For semantic-consistency loss, it keeps the structure of language embedding like word2vec.

prototypes independently but ignore explicitly considering the other pixels' relationship between them. To address it, we proposed to use the contrastive loss.

**Pixel-text score map:** In our framework, we use the score map to reduce the seen bias problem in ZS3. As shown in Fig.4, to get the discriminative joint embedding space, we compute the pixel-text score maps using the language prototypes $\mu_c \in \mathbb{R}^{k \times c}$ and the final outputs of feature maps $\upsilon \in \mathbb{R}^{h \times w \times c}$ by:

$$s = \hat{\upsilon}\hat{\mu}_c^T, s \in \mathbb{R}^{h \times w \times k} \tag{6}$$

where $\hat{\mu}_c$ and $\hat{\upsilon}$ are the $l_2$ normalized version of $\upsilon$ and $\mu_c$ along the channel dimension. By the way, the $\mu_c$ in the score map must use the seen classes prototypes only. Otherwise, it will make the unseen bias problem more serious. The score maps characterize the results of visual features pixel and language-guided semantic prototypes matching, which is one of the most crucial parts of our SwinZS3. First, we use the score maps to compute an auxiliary segmentation loss:

$$l_{aux} = CrossEntropy(Softmax(s/\tau), y) \tag{7}$$

where $\tau$ is a temperature coefficient which we set $0.07$ and $y$ is the ground truth label. The auxiliary segmentation loss can make the joint embedding space more discriminative which is beneficial to zero-shot semantic segmentation.

**Semantic consistency loss:** The semantic consistency loss $l_{sc}$ could transfer the relationship of $word2vec$ to the semantic prototypes embedding space. For adopting the pre-trained word embedding features, $l_{sc}$ could keep the class contextual information. The $l_{sc}$ defines the relation of prototypes as follows:

$$r_{ij}^{\mu} = \frac{e^{-\tau_\mu d(\mu_i, \mu_j)}}{\sum_{j \in S} e^{-\tau_\mu d(\mu_i, \mu_j)}} \tag{8}$$

where $d()$ is the distance between two prototypes and $\tau_\mu$ is a temperature parameter. Then for word embedding space, the relationship could be defined:

$$r_{ij} = \frac{e^{-\tau_s d(s_i, s_j)}}{\sum_{j \in S} e^{-\tau_s d(s_i, s_j)}} \tag{9}$$

So, the semantic consistency loss is defined as follows:

$$Lsc = -\sum_{i \in S} \sum_{j \in S} r_{ij} log \frac{r_{ij}^{\mu}}{r_{ij}} \tag{10}$$

The $l_{sc}$ could distill the word embedding contextual information to the prototypes.

### 3.5 NETWORK INFERENCE

During the inference, we use the semantic prototypes, which are the outputs of the semantic encoder as a NN classifier Cover & Hart (1967). We compute the euclidean distances and score maps from individual visual features to language prototypes, and classify each visual feature to the nearest language prototypes as follows:

$$\hat{y}(p) = \underset{c \in S \cup U}{argmin} \, d(\upsilon(p), \mu_c)(1 - sigmoid(s)) \tag{11}$$

where $d$ is the euclidean distance metric and $s$ is the score map. For the unseen classes $U$ still being biased towards the seen classes $S$, we adapted the work of Baek et al. (2021), which proposed the Apollonius circle. The $top2$ nearest language prototypes with individual visual features are $d_1$ and $d_2$. $d_1$ is the euclidean and score distance with the language prototype $\mu_1$ and $d_2$ is the distance with language prototype $\mu_2$ where we denote $c_1$ and $c_2$ as the class of $\mu_1$ and $\mu_2$. The decision rule is defined with the Apollonius circle as follows:

$$\hat{y}(p) = \begin{cases} c(p) & c_1 \in S \quad and \quad c_2 \in U \\ c_1 & otherwise \end{cases} \tag{12}$$

where

$$c(p) = c_1 \Pi[\frac{d_1}{d_2} \leq \gamma] + c_2 \Pi[\frac{d_1}{d_2} > \gamma] \tag{13}$$

We denote $\Pi$ as an function whose value 1 if the argument is true, and 0 otherwise. The $\gamma$ is an adjustable parameter which could modulate the decision boundary.

## 4 EXPERIMENTS

### 4.1 IMPLEMENTATION DETAILS

**Training:** For transformer backbone, we use the swin transformer (swin-tiny) proposed in Singh & Lee (2017), which is a baseline for transformer-baed ZS3 task. To avoid supervision leakage from unseen classes Xian et al. (2018), the backbone parameters are initialized with self-supervised model MoBY Xie et al. (2021b) pre-trained on Imagenet. We use an AdamW optimizer as the optimizer to train SwinZS3. For the backbone, we set the initial learning rate as $1 \times 10^{-4}$, and it uses the polynomial scheduler to decay at every iteration. The other parameters' learning rate is 10 times the backbone parameters' learning rate. The weight decay factor is set as $0.01$. For data augmentation, we keep the same setting with Baek et al. (2021). For other parameters $(\lambda, \gamma)$, we set $\lambda_1, \lambda_2$ to 0.1 and the $\gamma$ is 0.6.

**Dataset split:** We perform experiments on PASCAL VOC and PASCAL Context. The PASCAL-VOC2012 dataset contains 1464 training images and 1449 validation images with a total of 21 categories (20 object categories and background). The PASCAL Context dataset contains 4998 training and 5105 validation samples of 60 classes with 59 different categories and a single background class. Following the common practice, we adopt the 10582 augmented training samples for PASCAL VOC. For the zero-shot semantic segmentation network, we divide Pascal-VOC2012 training samples according to N-seen and 20-N unseen classes. For example, we choose cow and motorbike as unseen categories. Then we filter out those samples with cow and motorbike labels and train the segmentation network using the remaining samples. During the training time, the segmentation model should keep the $mIOU$ of unseen classes 0. We follow the experiment settings provided by ZS3Net which dividing the Pascal-VOC 2012 training samples 20 object classes into four splits (1) 18-2 classes (*cow*, *motorbike*), (2) 16-4 classes (*cat*, *sofa*), (3) 14-6 classes (*boat*, *fence*), and

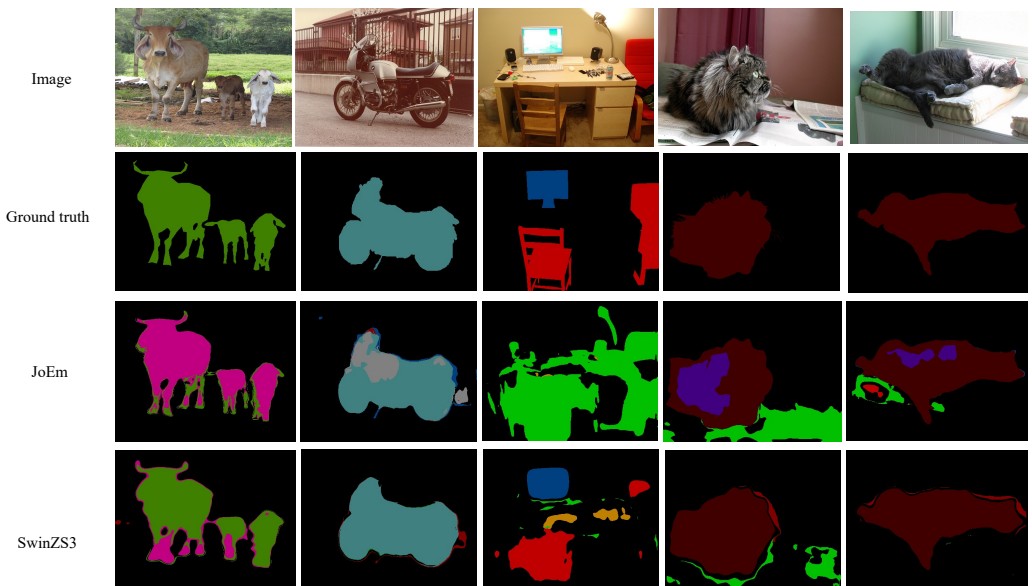

Figure 5: Qualitative results on PASCAL VOC. The unseen classes is "cow","motobike","cat". We compare the results of the other state-of-art method and our SwinZS3.

Table 1: Ablation study on the unseen-6 split of PASCAL Context by comparing $mIoU$ scores using different loss terms.

| method | $l_{ce}$ | $l_r$ | $l_{sc}$ | $l_{aux}$ | $mIoU_s$ | $mIoU_u$ | $hIoU$ |
|---|---|---|---|---|---|---|---|
| Deeplabv3+ | ✓ | | ✓ | ✓ | 33.4 | 8.4 | 13.4 |
| Deeplabv3+ | ✓ | ✓ | ✓ | | 36.2 | 23.2 | 28.3 |
| Deeplabv3+ | ✓ | ✓ | ✓ | ✓ | 37.7 | 25.0 | 30.2 |
| SwinZS3 | ✓ | | ✓ | ✓ | 25.8 | 12.0 | 16.4 |
| SwinZS3 | ✓ | ✓ | ✓ | | 37.1 | 24.3 | 29.3 |
| SwinZS3 | ✓ | ✓ | ✓ | ✓ | 39.3 | 26.2 | 31.4 |

(4) 12-8 classes (*bird,tvmonitor*). Each split contains previous unseen classes gradually. Then the model is evaluated on the full 1449 validation images.

**Evaluation metrics**: We use the mean intersection-over-union ($mIoU$) as evaluation metrics Long et al. (2015). In detail, we separately count the metrics of the seen classes and the unseen classes which are denoted by $mIoU_s$ and $mIoU_u$. We also adopt the harmonic mean ($hIoU$) of $mIoU_s$ and $mIoU_u$ for the arithmetic mean might be dominated by $mIoU_s$.

## 4.2 ABLATION EXPERIMENT AND RESULTS

**Ablation study:** In the table 1, we present an ablation analysis on two aspects: (a) CNNs vs Transformer (b) Whether to use the score map ($l_{aux}$) to modulate the decision boundary. For cross-entropy loss and regression loss are crucial to recognize the unseen classes, we choose the Deeplabv3+ with $l_{ce}$, $l_r$ and $l_{sc}$ as our baseline. For the first row, we report the baseline $IoU$ scores without $l_{aux}$. Then, we compare swin transformer (swin-tiny) backbone with Deeplabv3+ and transformer backbone gives gain of $hIoU$ 1.0 over the baseline. The second and third rows show that the $l_{aux}$ respectively give gain of $mIoU_u$ 3.0 and 3.1 over the baseline and 1.9, 2.1 over the SwinZS3 baseline. This shows a significant improvement for the ZS3 and gives a demonstration of the effectiveness of the two methods. Finally, we combine the transformer and score maps, and report the best $mIoU$ scores.

**Comparison to state-of-the-Arts:** As showed in table 2, we compare our approach with other state-of-art methods on PASCAL VOC and Context. We report the best scores of different split

Table 2: Quantitative results on the PASCAL VOC and Context validation sets. The numbers in bold are the best performance.

| K | method | VOC | | | Context | | |
|---|--------|------------|------------|------------|------------|------------|------------|
| | | $mIoU_s$ | $mIoU_u$ | $hIoU$ | $mIoU_s$ | $mIoU_u$ | $hIoU$ |
| 2 | DeViSE | 68.1 | 3.2 | 6.1 | 35.8 | 2.7 | 5.0 |
| | SPNet | 71.8 | 34.7 | 46.8 | 38.2 | 16.7 | 23.2 |
| | ZS3Net | 72.0 | 35.4 | 47.5 | 41.6 | 21.6 | 28.4 |
| | CSRL | 73.4 | 45.7 | 56.3 | 41.9 | 27.8 | 33.4 |
| | JoEm | 68.9 | 43.2 | 53.1 | 38.2 | 32.9 | 35.3 |
| | Ours | 69.2 | **45.8**(+2.6) | 55.3 | 39.8 | **33.5**(+(0.6)) | **36.3**(+(1.0)) |
| 4 | DeViSE | 64.3 | 2.9 | 5.5 | 33.4 | 2.5 | 4.7 |
| | SPNet | 67.3 | 21.8 | 32.9 | 36.3 | 18.1 | 24.2 |
| | ZS3Net | 66.4 | 23.2 | 34.4 | 37.2 | 24.9 | 29.8 |
| | CSRL | 69.8 | 31.7 | 43.6 | 39.8 | 23.9 | 29.9 |
| | JoEm | 67.0 | 33.4 | 44.6 | 36.9 | 30.7 | 33.5 |
| | Ours | 68.9 | **34.4**(+1.0) | **45.7**(+1.1) | 38.7 | **33.5**(+2.8) | **35.1**(+1.6) |
| 6 | DeViSE | 39.8 | 2.7 | 5.1 | 31.9 | 2.1 | 3.9 |
| | SPNet | 64.5 | 20.1 | 30.6 | 31.9 | 19.9 | 24.5 |
| | ZS3Net | 47.3 | 24.2 | 32.0 | 32.1 | 20.7 | 25.2 |
| | CSRL | 66.2 | 29.4 | 40.7 | 35.5 | 22.0 | 27.2 |
| | JoEm | 63.2 | 30.5 | 41.1 | 36.2 | 23.2 | 28.3 |
| | Ours | 62.6 | **31.6**(+1.1) | **42.0**(+0.9) | **39.3**(+3.1) | **26.2**(+3.0) | **31.4**(+3.1) |
| 8 | DeViSE | 35.7 | 2.0 | 3.8 | 22.0 | 1.7 | 3.2 |
| | SPNet | 61.2 | 19.9 | 30.0 | 28.6 | 14.3 | 19.1 |
| | ZS3Net | 29.2 | 22.9 | 25.7 | 20.9 | 16.0 | 18.1 |
| | CSRL | 62.4 | 26.9 | 37.6 | 31.7 | 18.1 | 23.0 |
| | JoEm | 58.5 | 29.0 | 38.8 | 32.4 | 20.2 | 24.9 |
| | Ours | 60.2 | **29.6**(+0.6) | **39.9**(+1.1) | **35.0**(+2.6) | **21.4**(+1.2) | **26.6**(+1.7) |

settings. And the other reported $mIoU$ are from Baek et al. (2021). For PASCAL Context dataset, the comparison shows that : we outperform the second best method JoEm by large margins, for the 6-split setting, we outperform 3.0 in $mIoU_u$ and 3.1 in $hIoU$. It is remarkably outperforming for the ZS3. We outperform the best generative ZS3 method GSRL Li et al. (2020) and give a $mIoU_u$ gain of 4.4 and $hIoU$ gain of 4.2, which shows the effectiveness and convenience of the discriminative method. And the CSRL has to be retrained when the novel unseen classes are added. However, our framework adopts one-stage training strategy (2) We achieve state-of-the-art performance on almost all the zero-shot settings like unseen-2,4,6,8 splits on $mIoU_u$ and $hIoU$. It confirms our approach could learn the discriminative representations. The PASCAL VOC dataset experiments emphasize the above analysis. The PASCAL VOC experiment shows a competitive performance.

**Qualitative results**. Fig.5 reports some qualitative examples from PASCAL VOC, with SwinZS3 modeling the unseen classes more accurately than its competitors. We notice that SwinZS3 can effectively reduce the false positive prediction. reports some qualitative examples from PASCAL VOC, with SwinZS3 modeling the unseen classes more accurately than its competitors.

## 5 CONCLUSION

We have proposed a transformer-based framework that exploits the visual and language features on the joint embedding space for zero-shot semantic segmentation. We have proposed to use the language-guided score map to better learn the discriminative space while innovatively modifying the decision boundary to reduce the seen bias problem. Finally, we experiment with our approach on standard ZS3 benchmarks and achieve a new state-of-the-art performance.

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
