# OpenReview forum: "SwinZS3: Zero-Shot Semantic Segmentation with a Swin Transformer"
_ICLR.cc/2023/Conference — Submitted to ICLR 2023_

### Official Review · Reviewer_nqEw · 2022-10-16

**Confidence:** 3
**Correctness:** 2
**Technical Novelty And Significance:** 1
**Empirical Novelty And Significance:** 1
**Recommendation:** 1

**Clarity, Quality, Novelty And Reproducibility:**

The clarity, quality and originality of the paper is relatively poor.
For detailed explanation, please see Cons. above.

**Strength And Weaknesses:**

Pros:
1. The topic of zero-shot semantic segmentation is interesting and important.
2. It achieves good benchmark results on Pascal VOC and Context datasets. (however the comparisons might be unfair, which will be described in Cons.)


Cons:
1. The novelty of this paper is limited. It seems that the paper claims its novelty as using Swin Transformer, however, Swin Transformer has been widely used in CV community and simply adopts it as the backbone is not novel. In addition, the paper describes the cross-entropy loss, regression loss, semantic consistency loss in details. However, these techniques are proposed by JoEm (Baek et al. ICCV'2021). This paper simply uses them without any modifications. For L_aux, many papers have used similar losses. for example, Dong, Xiaoyi, et al. "MaskCLIP: Masked Self-Distillation Advances Contrastive Language-Image Pretraining." arXiv preprint arXiv:2208.12262 (2022)..

2. The comparisons of the experiments (Table. 2) might be unfair. It uses stronger backbone (Swin Transformer) network compared to other methods. Please add experiments of DeepLabV3+ for all K in Table. 2.

3. Please also discuss the relationship with CLIP-based zero-shot segmentation methods in the related work section.

4. The paper writing needs significant improvement and careful revision. (1) The use of symbols is inconsistent. Does s in Eq.4 and Eq.6 the same? Some times N is used for the number of seen categories, sometimes K is used. (2) The introduction of "regression loss" is mostly unclear. where does semantic feature maps (s) come from? (3) The use of citation is wrong throughout the whole paper. (4) There are many typos. for example, extra "()" in Table. 2; "deeplabv3+" and "Deeplabv3+" in Table. 1.

Questions:
The paper writing is not clear enough.
1. In Table.1, Does the 1st row (deeplabv3+) mean JoEm?


**Summary Of The Paper:**

This paper proposes to tackle the problem of zero-shot semantic segmentation (ZS3). To capture the global feature relations and semantic information, SWIN transformer is adopted as the backbone. To improve the decision boundary, pixel-text auxiliary segmentation loss is used. Good benchmark results are obtained in Pascal VOC and Context datasets.

**Summary Of The Review:**

Due to lack of novelty, unfair experimental comparisons, and poor paper writing, the reviewer suggests rejecting this paper in its current form. The paper needs major revision before resubmitting.

---

> ### Author Response · Authors · 2022-11-17
> **Response to Reviewer nqEw**
>
> Q1:. It seems that the paper claims its novelty as using Swin Transformer, however, Swin Transformer has been widely used in CV community, and simply adopting it as the backbone is not novel.
>
> A1: Actually, it is not easy to use the swin transformer to ZS3. There are at least three questions to get good reasonable and good results.
>
> (1) For the pixel-text feature aligning work, what the network should be modified? For this, we did modify the swin-transformer pooling layer from avg pooling to max pooling. Because the avg pooling will cause the semantic shift problem, and we give a visual results for explaining this problem. For the zero-shot pixel-text framework, the avgpooling will cause the forground feature and background feature shift. And we think the maxpooling could alleviate this problem.
>
>
> (2) How to pretrain the swin-transformer?
> 	There are at least three chooses. 1. Using the common Imagenet pretrained weight.  It is not good for the supervision leakage. 2 Using the Imagnet dataset and removing the unseen classes labeled images. 3 Using the self-supervised weight. We did a lot of experiment for choosing the self-supervised weight.
>
> (3) The super-parameters setting.
>
>
> So, it is a basic but necessary work for adopting the swin-transformer to the zero-shot semantic segmentation. And the cross-entropy loss, regression loss, semantic consistency loss is not the key points in our paper, we never claim that this is our innovation. Though the pixel-score map is used in some works, the pixel-score decision boundary is never used in previous zero-shot works. We hope you could consider this job and your score more seriously. Thanks for you very much.
>
> Q2: The comparisons of the experiments (Table. 2) might be unfair. It uses stronger backbone (Swin Transformer) network compared to other methods. Please add experiments of DeepLabV3+ for all K in Table. 2.
>
> A2: In table.2 all the others use the Deeplabv3+ actually. The JoEm is one of  the typical approach for discriminative zero-shot semantic segmentation models using Deeplabv3+. And for the approach we proposed, the ablatin table.1 gives a resonable compare bettween Deeplabv3+ and swin-transformer.
>
> Q3:Please also discuss the relationship with CLIP-based zero-shot segmentation methods in the related work section.
>
>
> A3: We mentioned the CLIP-based methods in our introduction. Actually, we argue that the CLIP-based methods are not really zero-shot. Because we have used the labeled images or object for training the CLIP models. But the zero-shot model should never see the image-class label. So, for word2vec ZSSS works, it is not necessary to cite the CLIP-based methods in related work.
>
> Q4: The paper writing needs significant improvement and careful revision. (1) The use of symbols is inconsistent. Does s in Eq.4 and Eq.6 the same? Some times N is used for the number of seen categories, sometimes K is used. (2) The introduction of "regression loss" is mostly unclear. where does semantic feature maps (s) come from? (3) The use of citation is wrong throughout the whole paper. (4) There are many typos. for example, extra "()" in Table. 2; "deeplabv3+" and "Deeplabv3+" in Table. 1.
> A4: We have carefully proofread the article and have made the corresponding revision and make sure the use of symbols is consistent.

---

> > ### Comment · Reviewer_nqEw · 2022-12-10
> > **Thanks for the response**
> >
> > Thanks for your response. However, my major concerns are still not well addressed. So I tend to keep my original ratings.
> > 1. About the novelty of this paper. Thanks for your detailed response. I understand the difficulties of adapting Swin Transformer in the task of zero-shot segmentation. However, personally speaking, these modifications (e.g. avg pooling to max pooling,  pre-training, super-parameter setting) still lack technical novelties.
> > 2. About the fairness. I understand that all other compared models use Deeplabv3+, and the proposed model use Swin Transformer. And according to Table1, Swin-transformer achieves better performance than Deeplabv3+. That's why I say it might be unfair.
> > 3. About reference. I know you mentioned CLIP in introduction. And also there is a subsection ("Visual-language learning") in your related works. But why not discuss CLIP-based zero-shot segmentation works in the section of "Visual-language learning"? Aren't they more related to your work than CLIP/ALIGN?

---

### Official Review · Reviewer_S84P · 2022-10-24

**Confidence:** 4
**Correctness:** 3
**Technical Novelty And Significance:** 2
**Empirical Novelty And Significance:** 2
**Recommendation:** 5

**Clarity, Quality, Novelty And Reproducibility:**

The overall quality of the article is acceptable. The essay approach is slightly lacking in novelty. There are problems with the presentation and writing of the article. The reproducibility is okay if the code is released.

**Strength And Weaknesses:**

Strength

      -The proposed method achieved relatively good results on benchmarks.


Weakness


      -The presentation of some parts of the article is unclear. For example I'm confused about the structure of Language Encoder, but the article doesn't explain what it is throughout.
      -There are many problems with the layout and writing of the article. For example, there is incorrect capitalization in the first line of section 3.1 and missing spaces in line 4. There is confusion about the case of symbols in section 3.4.
     -Some recent related papers are not cited. Such as [1, 2, 3]
    [1] https://arxiv.org/abs/2112.14757
    [2] https://openaccess.thecvf.com/content/CVPR2022/papers/Ding_Decoupling_Zero-Shot_Semantic_Segmentation_CVPR_2022_paper.pdf
    [3] https://arxiv.org/abs/2202.11094

**Summary Of The Paper:**

The article proposed a new network SwinZS3 that applies swin transformer to the zero-shot semantic segmentation domain. Experimental results on PASCAL VOC and PASCAL Context show that the method proposed in this paper outperforms existing methods.

**Summary Of The Review:**

The method proposed in the article is not novel enough. In addition, there are major problems with the writing and layout of the article. I deem that it's not a solid work. So I tend to give weak reject.

---

> ### Author Response · Authors · 2022-11-17
> **Respond to the Reviewer S84P**
>
> Q1: The structure of Language Encoder.
>
> A1: The Language Encoder which we use in paper is just an MLP layer. We have add this explanation on our paper. Thanks for your advice!
>
> Q2: There are many problems with the layout and writing of the article. For example, there is incorrect capitalization in the first line of section 3.1 and missing spaces in line 4. There is confusion about the case of symbols in section 3.4.
>
> A2: We have carefully proofread the article and have made the corresponding revision and make sure the use of symbols is consistent.
>
> Q3: Some recent related papers are not cited. Such as [1, 2, 3]\\
>
> A3: These papers are cited in the new version. Thanks!
>
> Q4: Due to lack of novelty, unfair experimental comparisons, and poor paper writing, the reviewer suggests rejecting this paper in its current form. The paper needs major revision before resubmitting.
>
> A4:
>  Actually, it is not easy to use the swin transformer to ZS3. There are at least three questions to get good reasonable and good results.
>
> (1) For the pixel-text feature aligning work, what the network should be modified?
>
> For this, we did modify the swin-transformer pooling layer from avg pooling to max pooling. Because the avg pooling will cause the semantic shift problem, and we give a visual results in the supplementary material for explaining this problem. For the zero-shot pixel-text framework, the avgpooling will cause the forground feature and background feature shift. And we think the maxpooling could alleviate this problem.
>
> (2) How to pretrain the swin-transformer?
>
> There are at least three chooses. 1. Using the common Imagenet pretrained weight. It is not good for the supervision leakage. 2 Using the Imagnet dataset and removing the unseen classes labeled images. 3 Using the self-supervised weight. We did a lot of experiment for choosing the self-supervised weight.
>
> (3) The super-parameters setting.
>
> So, it is a basic but necessary work for adopting the swin-transformer to the zero-shot semantic segmentation. And the cross-entropy loss, regression loss, semantic consistency loss is not the key points in our paper, we never claim that this is our innovation. Though the pixel-score map is used in some works, the pixel-score decision boundary is never used in previous zero-shot works. We hope you could consider this job and your score more seriously. Thanks for you very much.

---

### Official Review · Reviewer_xJ8c · 2022-10-25

**Confidence:** 4
**Correctness:** 2
**Technical Novelty And Significance:** 2
**Empirical Novelty And Significance:** Not applicable
**Recommendation:** 3

**Clarity, Quality, Novelty And Reproducibility:**


The work has limited novelty: there are already uses of transformer models for semantic segmentation problems, the architecture change itself is not very interesting, and the other contributions are mainly adaptations of existing ZSL/semantic segmentation formulations. Due to the reasons mentioned above, the study includes various deficiencies in reproducibility.

Importantly: it is not clear *how* the hyper-parameters were chosen. While several hyper-parameters are shared, it is not clear how they are tuned and whether unseen class samples (test examples) have been used in the process.

**Strength And Weaknesses:**

Strengths:
- The proposed transformer framework appears to be promising. The obtained limited experimental results (please see weaknesses) also support this situation.
- Nice to see the impact of auxiliary segmentation loss on SwinZS3 and DeepLab methods.
Weaknesses:
- The text flow feels irregular. For example, CLIP-based meyhods are mentioned regardless of the context at the end of the Introduction section. Some figures are not referred from the text (eg, Fig.1 and Fig.4). Grammatical problems also disrupts the flow. In addition, the text contains too many typos; e.g. "fro", "networkBaek", "generatining", etc.
- The authors follow the experimental settings provided by ZS3Net, and use 4 different splits for comparison with previous methods. However, there are 5 different splits according to the ZS3Net method and they share the results for 5 different splits in their paper. Hence, unseen-10 split results seem to be missing for both Pascal VOC and Pascal Context datasets.
- When the results shared in Table 2 are examined, it is observed that the CSRL method generally obtains better results for the seen classes. However, there is no clear experimental result to support the central claim that the seen/unseen bias is alleviated via minimizing the euclidean distance and using the pixel-text score maps.

*ZS3Net: Bucher, M., Vu, T. H., Cord, M., & Pérez, P. (2019). Zero-shot semantic segmentation. Advances in Neural Information Processing Systems

**Summary Of The Paper:**

In this paper, the authors propose a discriminative zero-shot semantic segmentation approach that uses transformer image and language encoders to exploit the global relations of visual features and to reduce seen class bias. In this approach, the image encoder is trained with calculating pixel-text score maps which use dense language-guided semantic prototypes. The authors argue that CNN-based ZS3 models sometimes fail to extract language-guided activation fields because they work in reduced receptive fields and use fewer attention mechanisms to extract global relations. Experimental results on benchmark Pascal VOC and Pascal Context datasets show that this approach (i.e. SwinZS3) obtains state-of-the-art results.

**Summary Of The Review:**

The arguments put forward by the authors are worth examining. (Limited) experiments also show that the method obtains successful results. However, the main and additional experiments are incomplete from my point of view. In addition, the study also needs serious improvements in writing, the overall manuscript quality does not meet the ICLR standards.

---

> ### Author Response · Authors · 2022-11-17
> **Response to Reviewer xJ8c**
>
> Q1: The text flow feels irregular.
>
> A1:  We have carefully proofread the article and have made the corresponding revision and make sure the use of symbols is consistent.
>
> Q2: Unseen-10 split results seem to be missing for both Pascal VOC and Pascal Context datasets.
>
> A2: We put the results in the supplementary material rebuttal pdf. Due to papes limitations, we did not put the results in the camera-ready paper. Though the Unseen-2 -8 split results are enough to illustrate its performance, we consider adjusting the paper for adding the Unseen-10 split results. Thanks for your advice.
>
> Q3：The work has limited novelty: there are already uses of transformer models for semantic segmentation problems, the architecture change itself is not very interesting, and the other contributions are mainly adaptations of existing ZSL/semantic segmentation formulations. Due to the reasons mentioned above, the study includes various deficiencies in reproducibility.
>
> A3:
> Actually, it is not easy to use the swin transformer to ZS3. There are at least three questions to get good reasonable and good results.
>
> (1) For the pixel-text feature aligning work, what the network should be modified?
>
> For this, we did modify the swin-transformer pooling layer from avg pooling to max pooling. Because the avg pooling will cause the semantic shift problem, and we give a visual results in the supplementary material  for explaining this problem. For the zero-shot pixel-text framework, the avgpooling will cause the forground feature and background feature shift. And we think the maxpooling could alleviate this problem.
>
> (2) How to pretrain the swin-transformer?
>
> There are at least three chooses. 1. Using the common Imagenet pretrained weight.  It is not good for the supervision leakage. 2 Using the Imagnet dataset and removing the unseen classes labeled images. 3 Using the self-supervised weight. We did a lot of experiment for choosing the self-supervised weight.
>
> (3) The super-parameters setting.

---

> > ### Author Response · Authors · 2022-11-17
> > **Respond to Review xj8c**
> >
> > So, it is a basic but necessary work for adopting the swin-transformer to the zero-shot semantic segmentation. And the cross-entropy loss, regression loss, semantic consistency loss is not the key points in our paper, we never claim that this is our innovation. Though the pixel-score map is used in some works, the pixel-score decision boundary is never used in previous zero-shot works. We hope you could consider this job and your score more seriously. Thanks for you very much.

---

### Official Review · Reviewer_sxBw · 2022-10-28

**Confidence:** 4
**Correctness:** 3
**Technical Novelty And Significance:** 2
**Empirical Novelty And Significance:** 2
**Recommendation:** 5

**Clarity, Quality, Novelty And Reproducibility:**




Introduction:
WSSS are often based on easily obtaining annotations, such as scribbles => sentence seems incorrect

We argue that a shared shortcoming of previous ZS3 models falls in the reduced receptive field of CNNs and less uses attention mechanisms for extracting the global relations of visual features conditioned with language semantic information. => 'less uses' does not seem right

Current networkBaek et al. (2021) adopt traditional => space between network and Baek. Sentence also is not correct

Although generative methods achieve impressive performance in zero-shot semantic segmentation tasks.The methods are limited by a multi-stage training strategy, Remove although or replace '.' with ','

The lsc is proposed by Baek et al. (2021), which define the relation between prototypes as follows: => language does not seem correct

The overall loss is finally formulated as
L = Lce + Lr + λ1Lsc + λ2Laux, what is Laux in equation 2, I only see Lps in equation 2



**Strength And Weaknesses:**

The method obtains reasonable improvements over good baselines and state of the art results.
The use of pixel text score map is interesting.


No ablation experiments for Lsc and Lr, which are described as sub-sections of the technical contribution in the approach.
The paper is not written well and has several errors.

**Summary Of The Paper:**

The paper proposes a transformer based approach for zero shot semantic segmentation. It makes the use of different loss functions like cross entropy loss for seen classes, regression loss between language and visual features to account for unseen classes, a pixel text score map to reduce the seen bias problem and a semantic consistency loss to transfer the relationship of word2vec features to the semantic prototypes of the embedding space. State of the art results are shown on Pascal VOC and Context datasets.

**Summary Of The Review:**

The paper has many errors and has a few missing ablation experiments. The method does not seem very novel as it mixes a combination of loss functions (mostly known) with transformers to improve results marginally over baselines. Overall the contributions are borderline and near the acceptance threshold if some concerns are alleviated.

Updated rating after reading other reviews.

---

> ### Author Response · Authors · 2022-11-17
> **Response to Reviewer sxBw**
>
> Q1: Abut the English grammar mistakes in the paper.
>
> A1: Thanks for your correction. We have carefully proofread the article and have made the corresponding revision and make sure the use of symbols is consistent.
>
> Q2: No ablation experiments for Lsc and Lr, which are described as sub-sections of the technical contribution in the approach.
>
> A2: We have add the ablation experiments for Lsc and Lr for Deeplabv3+ and SwinZS3 in table1. Thanks for your advice.

---

### Decision · Program_Chairs · 2023-01-20

**Decision:**

Reject

**Justification For Why Not Higher Score:**

Four knowledgeable reviewers recommend rejection. There is a rebuttal, and during the discussion, the reviewers reached a consensus that the paper has merits but is not ready for publication as there are several concerns in the paper's writings,  claims that lack sufficient support and design choices that lack sufficient justification, and missing experiments. The paper does not give a clear and strong intuition why the method work. No basis for overturning the reviews.

**Justification For Why Not Lower Score:**

N/A.

**Metareview: Summary, Strengths And Weaknesses:**

The paper proposes a transformer-based approach for zero-shot semantic segmentation. To improve the decision boundary, pixel-text auxiliary segmentation loss is used and was shown to improve the performance.

Strengths
-------------


- Reviewer sxBw, xJ8c,S84PJ,r nqEw: The method obtains reasonable improvements over good baselines and state-of-the-art results. Reviewer xJ8c: Nice to see the impact of auxiliary segmentation loss on SwinZS3 and DeepLab methods.
- Reviewer xJ8c: The proposed transformer framework appears to be promising.
- Reviewer nqEw: The topic of zero-shot semantic segmentation is interesting and important.

Weaknesses
-------------

- Reviewer sxBw: No ablation experiments for Lsc and Lr, which are described as sub-sections of the technical contribution in the approach.
- Reviewer sxBw, xJ8c, S84P, nqEw : The paper is not written well and has several grammatical errors. Detailed comments by reviewers clarify the details.
- Reviewer xJ8c: there is no clear experimental result to support the central claim that the seen/unseen bias is alleviated via minimizing the euclidean distance and using the pixel-text score maps.
- Reviewer xJ8c:  missing experiments for both Pascal VOC and Pascal Context datasets.
-Reviewer nqEw : The novelty of this paper is limited. It seems that the paper claims its novelty as using Swin Transformer and compared to MaskCLIP paper (Dong, Xiaoyi, et al. ").
- The comparisons of the experiments (Table. 2) might be unfair. It uses a stronger backbone (Swin Transformer) network compared to other methods. Please add experiments of DeepLabV3+ for all K in Table. 2.
- Missing discussion and experimental depth.


In conclusion, all reviewers are leaning towards not accepting the paper with its current version.



**Summary Of Ac-Reviewer Meeting:**

N/A.